# Understanding the Actual Use of Anti-HIV Drugs in Japan from 2016 to 2019: Demonstrating Epidemiological Relevance of NDB Open Data Japan for Understanding Japanese Medical Care

**DOI:** 10.3390/ijerph191912130

**Published:** 2022-09-25

**Authors:** Hiroyuki Tanaka, Toshihisa Onoda, Toshihiro Ishii

**Affiliations:** Department of Practical Pharmacy, Faculty of Pharmaceutical Sciences, Toho University, 2-2-1 Miyama, Funabashi, Chiba 274-8510, Japan

**Keywords:** NDB Open Data Japan, anti-HIV drug, continuous survey, tenofovir alafenamide fumarate, integrase strand transfer inhibitors, single-tablet regimen

## Abstract

The National Database of Health Insurance Claims and Specific Health Checkups of Japan (NDB) Open Data Japan is helpful for attaining simple and comprehensive understanding of medical care in Japan. Herein, we investigated the transition of anti-HIV-drug use in Japan over a 4-year period from fiscal year (FY) 2016 to FY 2019 using data on anti-HIV drugs that were extracted from the 3rd, 4th, 5th, and 6th NDB Open Data Japan. Then, the data were stratified by mechanism of action, single-tablet regimen (STR) or non-STR, age groups, and sex and analyzed. Throughout the study period, the prescription volume for tenofovir alafenamide fumarate as the backbone drug and integrase strand transfer inhibitors as the anchor drug increased. In FY 2019, STRs constituted approximately 44% of the total combination antiretroviral therapy regimens, 1.6 times higher than that in FY 2016 (27%). With the advent of newer drugs and regimens, the differences in anti-HIV drugs prescribed to patients of different ages and sex gradually diminished; however, differences were unremarkable in the first period, especially between sexes. The NDB Open Data Japan made it relatively easy to evaluate recent trends in anti-HIV prescription in Japan, indicating its usefulness for continuous surveys in this field.

## 1. Introduction

Anti-human immunodeficiency virus (anti-HIV) drugs are being used to treat people living with HIV (PLWH) globally. Since 1997, combination antiretroviral therapy (cART) with three or more anti-HIV drugs recommended by national and international HIV treatment guidelines has been available, making it possible to strongly suppress viral proliferation and restore immunity [1,2]. The early cART regimen had many limitations, such as various adverse events, drug interactions with concomitant medication or food, the need to consume more tablets, and the need to consume medication several times a day [3]. Thereafter, newer anti-HIV drugs with relatively few adverse effects, less interaction with concomitant drugs, and longer half-lives were developed, leading to the development of more effective cART [3]. Among antiviral drugs, the development of anti-HIV drugs is particularly remarkable [4]; therefore, international and national HIV treatment guidelines are updated frequently. The improvement in quality of cART combined with the development of newer anti-HIV drugs has improved the quality of life of PLWH and decreased morbidity and mortality. Life expectancy for PLWH is now approximately similar to that of non-HIV-infected individuals [5]. Therefore, in recent years, aging of PLWH has also become an important consideration for healthcare professionals involved in the treatment of HIV infection [6]. Because some anti-HIV drugs are excreted by the kidneys, they may cause damage to the kidneys, while some anti-HIV drugs affect bone, glucose, and lipid metabolism [7]. In addition, older PLWH often have multiple comorbidities, with subsequent polypharmacy being a risk factor for drug interactions [8]. Thus, the age of PLWH may influence anti-HIV-drug selection.

Five classes (by mechanism of action) of anti-HIV drugs are used in Japan: nucleoside reverse transcriptase inhibitor (NRTI), non-nucleoside reverse transcriptase inhibitor (NNRTI), protease inhibitor (PI), integrase strand transfer inhibitor (INSTI), and C-C chemokine receptor type 5 (CCR5) antagonist [2]. In many cases, HIV treatment guidelines recommend selecting a cART regimen, which consists of two NRTIs as backbone drugs plus a third “anchor drug” from another drug class (INSTIs, PIs, or NNRTIs), at the start of cART [1,2]. In addition, PLWH sometimes have cART-regimen changes in clinical practice [3,9]. Regimen changes in cART may occur due to the thought that the use of newer anti-HIV drugs may lead to better clinical outcomes or because patients may be dissatisfied with their current cART regimen [3]. Adverse effects, drug interactions, poor adherence, or simplification of anti-HIV drugs with a single-tablet regimen (STR) may also prompt cART-regimen changes [3,8]. As mentioned previously, due to frequent revisions of guidelines and the aging of PLWH, the real-world use of anti-HIV drugs is likely to be more diverse. However, to date, there are only a limited number of reports that examine the actual usage and trends of usage of anti-HIV drugs in Japan [9].

The National Database of Health Insurance Claims and Specific Health Checkups of Japan (NDB) is an insurance claim database that collects health insurance claims since 2009 and specific health checkup/guidance data since 2008, resulting in one of the most exhaustive national healthcare databases in the world [10]. The NDB includes data for diagnosis, age, sex, admission and discharge dates, drug prescriptions, procedure performed, and health checkup data. Since 2011, the NDB data have also been available as secondary data for research [10]. Recently, epidemiological studies using NDB data to analyze the trend of cART-regimen changes, comorbidities, and concomitant medications in PLWH have been conducted [6,9]. However, due to the high confidentiality of the data, direct access to the data is limited to researchers who can ensure data security.

Recently, the Ministry of Health, Labor and Welfare (MHLW) published the “NDB Open Data Japan” online and provided several summary files based on the NDB data [10]. The NDB Open Data Japan (including data on prescription volume from 1 April 2014 to 31 March 2015) was first published in October 2016 and has been published every year since then. The NDB Open Data Japan can be used for simple and comprehensive understanding of medical care in Japan [10], and the number of studies using data from the NDB Open Data Japan has been increasing [11]. Moreover, our previous study has suggested that research using the NDB Open Data Japan is useful for identifying prescription trends of anti-HIV drugs and estimating the number of patients treated with cART [11]. If the usefulness of epidemiological surveys using the NDB Open Data Japan for continuous surveys on the actual use of anti-HIV drugs is demonstrated, updating information related to this field will become easier.

Therefore, in this study, we investigated the transition of anti-HIV drug use in Japan over a four-year period from fiscal year (FY) 2016 to FY 2019 using the NDB Open Data Japan. Moreover, the usefulness of the NDB Open Data Japan for continuous surveys in this field was discussed by comparing the results of this study with previously reported analysis findings of the NDB data.

## 2. Materials and Methods

### 2.1. Data Source

The authors obtained the 3rd, 4th, 5th, and 6th NDB Open Data Japan from the website of MHLW (https://www.mhlw.go.jp/stf/seisakunitsuite/bunya/0000177182.html, accessed on 31 October 2021); the collection periods for these data were FY 2016 (1 April 2016, to 31 March 2017), FY 2017 (1 April 2017, to 31 March 2018), FY 2018 (1 April 2018, to 31 March 2019), and FY 2019 (1 April 2019, to 31 March 2020), respectively. The NDB Open Data Japan provides data on 100 products in descending order of prescription volume in each efficacy category, in which the low frequency products have been excluded [10]. Additionally, the actual prescription volume of oral drugs with prescription volume ≤ 1000 was anonymized [10]. There were two spreadsheet files regarding prescribed oral drugs: one was categorized by sex and 5-year age groups, and the other was categorized by 47 prefectures in Japan [10]. In this study, we extracted the records on anti-HIV drugs prescribed in- or out-of-hospital to inpatients and outpatients.

### 2.2. Backbone Drug and Anchor Drug

According to HIV treatment guidelines, a cART regimen generally consists of two NRTIs plus a third active anti-HIV drug from one of three drug classes (mechanism of action), INSTI, PI, or NNRTI. In this study, we considered two NRTIs as backbone drugs and the third active anti-HIV drug (INSTI, PI, or NNRTI) as an anchor drug [2]. Tenofovir disoproxil fumarate (TDF) + lamivudine (3TC) or emtricitabine (FTC), tenofovir alafenamide fumarate (TAF) + 3TC or FTC, and abacavir (ABC) + 3TC were recommended as two NRTIs in the HIV treatment guidelines during the study period. Therefore, in this study, we investigated the usage trends of TDF, TAF, and ABC as backbone drugs and INSTIs, PIs, and NNRTIs (including individual drugs belonging to each drug class) as anchor drugs.

### 2.3. Data Processing and Analysis

The NDB Open Data Japan data are summarized by product name, without information regarding the ingredients. Some anti-HIV drugs currently in use have a combination of multiple ingredients with different mode of action. Therefore, data were processed after converting product names to ingredient names. Data were processed based on previous methods [11]. In brief, the number of patients was used to compare the prescription trends for each anti-HIV drug. Specifically, the number of patients represented here is the prescription volume of each anti-HIV drug divided by 365 and the daily dose stated in the Japanese package insert. These data were stratified by mechanism of action of anti-HIV drugs (NRTI, NNRTI, PI, or INSTI), STR or non-STR, age groups (20–49 or 50+ years), and sex (male or female) and used for further analysis. Demographic characteristics and anti-HIV-drug prescriptions during the study period were summarized descriptively using the number and percentage of patients for categorical variables. All statistical analyses were performed using Microsoft^®^ Excel^®^ 2016 (Microsoft Corp, Redmond, WA, USA).

### 2.4. Ethics

The NDB Open Data Japan comprises anonymized open data and does not include individual patient information. Thus, ethical review and informed consent were not required.

## 3. Results

### 3.1. Anti-HIV Drugs Included in This Study and Their Characteristics

In the NDB Open Data Japan, anti-HIV drugs belonged to the efficacy category of antiviral drugs. In the 3rd, 4th, 5th, and 6th Open Data Japan, anti-HIV drugs included in the top 100 prescribed antiviral drugs are shown in Table 1. The 3rd, 4th, 5th, and 6th NDB Open Data Japan contained 27, 25, 24, and 23 products, respectively, which were used to treat HIV infections. In addition, Table 1 includes information on the ingredients of each product, mechanisms of action, and STR or non-STR.

### 3.2. Distribution of Backbone Drug Prescriptions by Year

During the study period, the estimated number of patients treated with TDF, TAF, or ABC increased from 18,521 to 21,996 (Figure 1). 3TC or FTC are used in combination with TDF, TAF, ABC, and other backbone drugs making them the most commonly used NRTIs. The estimated number of patients calculated based on 3TC or FTC usage throughout the study period was 18,673 and 22,157 in FY 2016 and FY 2019, respectively. There was very little difference (<1%) in the estimated number of patients receiving backbone drugs based on the two indicators. In Japan, the prescription volume of TAF in FY 2016 was small, since TAF was only available for prescription from January 2017. Thereafter, in FY 2017, the tenofovir products rapidly replaced TDF with TAF; and in FY 2019, TAF prescription comprised 67.7% of all backbone drug prescriptions (more than 95% of the tenofovir products).

### 3.3. Distribution of Anchor Drug Prescriptions by Year (Figure 2)

During the study period, the estimated number of patients treated with INSTIs, PIs, or NNRTIs increased from 20,391 to 22,826. INSTIs accounted for approximately 65.6% of all anchor drugs prescribed in FY 2016. This percentage gradually increased throughout the study period, reaching 84% in FY 2019. Among INSTIs, the volume and percentage of dolutegravir (DTG) and elvitegravir (EVG) prescriptions increased from FY 2016 to FY 2018. However, once bictegravir (BIC) was available for prescription from April 2019, the volume and percentage of DTG and EVG prescriptions began to decline. The volume and percentage of PI prescriptions declined throughout the study period. In particular, atazanavir, fosamprenavir, and nelfinavir were not included in the 6th NDB Open Data Japan (FY 2019). The volume and percentage of NNRTI prescriptions also declined throughout the study period. In particular, because rilpivirine was not included in the 6th NDB Open Data Japan (FY 2019), the apparent number of NNRTI users decreased considerably.

**Figure 2 ijerph-19-12130-f002:**
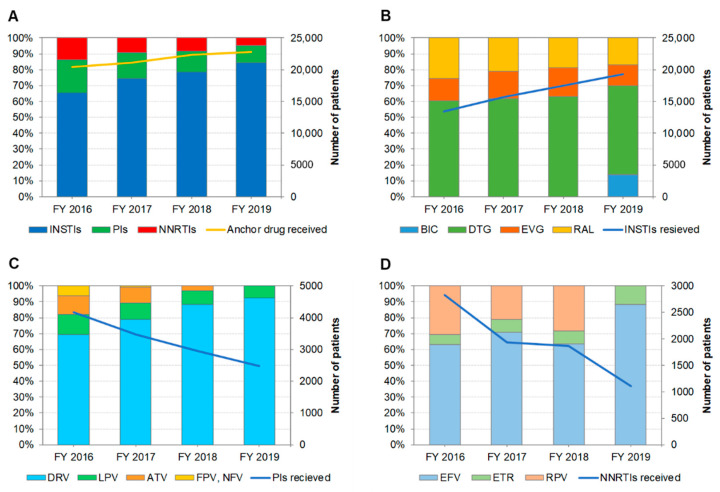
Changes in anchor drug prescriptions from FY 2016 to FY 2019. (**A**) Changes in prescription of class (mechanism of action) selected as anchor drug, (**B**) changes in prescription of individual INSTI drugs, (**C**) changes in prescription of individual PI drugs, (**D**) changes in prescription of individual NNRTIs drugs. The analysis was performed on drugs listed in Table 1. INSTI: integrase strand transfer inhibitor, NRTI: nucleoside reverse transcriptase inhibitor, NNRTI: non-nucleoside reverse transcriptase inhibitor, PI: protease inhibitor, BIC: bictegravir, DTG: dolutegravir, EVG: elvitegravir, RAL: raltegravir, DRV: darunavir, LPV: lopinavir, ATV: atazanavir, FPV: fosamprenavir, NFV: nelfinavir, EFV: efavirenz, ETR: etravirine, RPV: rilpivirine.

### 3.4. Distribution of STR Prescriptions by Year (Figure 3)

The number of patients treated with STR increased throughout the study period. In FY 2019, it was estimated that approximately 44% of patients receiving cART were treated with STRs, 1.6 times higher than that in FY 2016 (27%). Almost all patients treated with STR were prescribed INSTIs as anchor drugs. However, it should be noted that STR containing TAF as a backbone drug has been preferred in recent years.

**Figure 3 ijerph-19-12130-f003:**
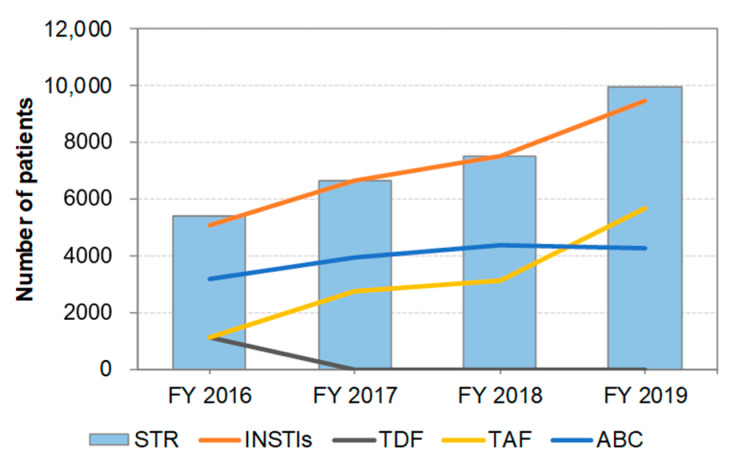
Change in the number of patients treated with STR from FY 2016 to FY 2019. The line graph indicates the number of patients treated with each ingredient of the STR (INSTIs, TDF, TAF, or ABC). The analysis was performed on drugs listed in Table 1. STR: single-tablet regimen, INSTI: integrase strand transfer inhibitor, TDF: tenofovir disoproxil fumarate, TAF: tenofovir alafenamide fumarate, ABC: abacavir.

### 3.5. Comparison of Anti-HIV-Drug Prescriptions by Age Group

Changes in the prescriptions of backbone drugs by age group are shown in Figure 4. In FY 2016, when TDF accounted for the majority of tenofovir prescriptions, the difference in percentage of tenofovir prescriptions between the 20–49 and 50+ age group was 14%. However, as TAF, the newer prodrug of tenofovir, was preferred over TDF, the difference in the percentage of tenofovir prescriptions between the two age groups decreased. The reason for this change was the increased use of TAF in the 50+ age group.

Changes in the prescriptions of anchor drugs by age group are shown in Figure 5. Throughout the study period, the percentage of INSTIs prescriptions increased, and the percentage of PI and NNRTI prescriptions decreased in both age groups. In each year, the percentage of prescriptions for PIs between the 20–49 and 50+ age group was similar.

The proportion of patients aged 50+ receiving cART estimated from backbone drug or anchor drug prescriptions gradually increased throughout the study period (Figure 4 and Figure 5).

### 3.6. Comparison of Anti-HIV-Drug Prescriptions by Sex

In Japan, males accounted for approximately 93% of patients receiving cART, and this proportion did not change throughout the study period (Figure 6 and Figure 7). Thus, the use of anti-HIV drugs in Japan may reflect the usage in the male population.

Changes in the prescriptions of backbone drugs and anchor drugs by sex are shown in Figure 6 and Figure 7, respectively. Changes in backbone drug use were very similar between males and females. For anchor drugs, the percentage of PIs used was slightly higher for females than for males in FY 2016, but this difference disappeared throughout the study period. Therefore, this study does not report any clear differences characterized by sex in the use of anti-HIV drugs in recent years.

## 4. Discussion

This study revealed the actual use of anti-HIV drugs from FY 2016 to FY 2019. By using the NDB Open Data Japan, it was relatively easy to identify changes in anti-HIV-drug use. Furthermore, it was possible to investigate changes in the STR prescription and differences in prescription according to age or sex. Recently, some epidemiological studies using the NDB were reported in which the study participants were PLWH [6,9]. The NDB is the largest nationwide cross-sectional database in Japan [10] and is extremely useful as a research resource to support clinical studies [12,13]. The NDB includes data for diagnosis, age, sex, admission and discharge dates, drug prescriptions, procedures performed, and health checkup data. However, the use of the NDB requires researchers to meet certain requirements, and the request must be reviewed by a panel of experts from the perspective of research content and security in the research environment. In addition, research using NDB also requires an ethical review at the researcher’s institution. In a survey that continuously updates information, as in our study, this procedure is cumbersome and unreasonable. Therefore, we focused on the NDB Open Data Japan, which does not have such requirements. The NDB Open Data Japan is available to everyone, but the data provided is summarized [10]. Therefore, when used by researchers, it is necessary to process the data based on their own preferences prior to analysis.

In this study, we aimed to understand changes in anti-HIV-drug prescriptions in Japan based on multi-year data. Since information on cART is not available from the NDB Open Data Japan, the number of patients receiving cART was estimated from the amount of backbone drug or anchor drug prescriptions. From NDB data analysis, Naito et al. reported that the numbers of patients who received cART in FY 2016, FY 2017, and FY 2018 were 20,419, 21,881, and 22,909, respectively [6]. Therefore, the number of patients estimated from anchor drug prescriptions in this study appears to be closer to the actual number of patients than that data derived from backbone drug prescriptions. Among backbone drugs, TDF, 3TC, and FTC are used at reduced doses in patients with chronic kidney disease and renal impairment [14]. NRTIs may also be unfeasible due to adverse drug effects, accumulated toxicity, or drug resistance. In such cases, NRTI-sparing cART regimens may be used [15]. These factors may explain why the number of patients estimated from anchor drug prescriptions was more accurate than that from backbone drug prescriptions. In addition, the number of patients who received cART in FY 2015, which we estimated by analyzing the 2nd NDB Open Data Japan in a previous study [11], was similar to the results of a nationwide questionnaire survey [16]. Thus, it was concluded that the analysis results of the NDB Open Data Japan had almost the same accuracy as the research using the NDB data and the questionnaire survey on a nationwide scale.

In this study, changes in the prescription of TDF, TAF, and ABC used as backbone drugs from FY 2016 to FY 2019 were studied. In January 2017, it became possible to use TAF, a new prodrug of tenofovir and a potential successor of TDF [17], in Japan. Although TDF is highly effective against HIV and also has therapeutic effects against HBV, there have been concerns about its renal and bone toxicity. Although ABC is less toxic to kidneys and bone than TDF, it has no therapeutic effect against HBV. TAF appears to be as effective as TDF against HIV or HBV, with lower renal and bone toxicity [17]. In addition, TAF does not require dose adjustment in patients with renal impairment [17]. Therefore, TAF can be used in patients with renal impairment or in older patients. Furthermore, the prescription volume of TAF is gradually increasing. Changes in prescribing trends for INSTIs, PIs, and NNRTIs used as anchor drugs were also revealed. Furthermore, changes in the prescribing trend of individual drugs included in three classes (mechanism of action) were characterized. Throughout the study period, the percentage of INSTI prescriptions consistently increased, while that of PIs and NNRTIs decreased. Despite INSTIs being equivalent to NNRTIs and PIs in terms of therapeutic effects against HIV, the increasing preference of INSTIs could be due to less adverse drug events [9] and drug interactions [8]. For individual drugs belonging to INSTIs, the drugs that can be incorporated in STRs and drugs that have fewer drug interactions are prioritized. Some studies suggest that adherence and persistence may be higher among patients treated with STRs compared with those treated with multi-tablet regimens [18,19,20]. Prescriptions for STR increased throughout the study period, and the rate of increase in FY 2019 was high compared to that in other years. This could be due to introduction of STR-compatible formulation that combines BIC as a newer INSTI with TAF + FTC as backbone drugs. In recent years, further advances have been made in terms of simplification of HIV treatment. Dual therapy regimens (DTG + 3TC or DTG + RPV) are usually indicated for patients with no previous virological failure and no resistance-associated mutations to ingredients of the said regimen and are selected to minimize adverse drug events, drug interactions, and pill burden [21]. Trends in the use of dual therapy regimens should also be investigated in the future.

Differences in age [22] and sex [23,24] create differences in body size, pharmacokinetics, and occurrence of adverse drug events. In women, the possibility of pregnancy is also taken into consideration depending on their age [23,24,25]. These factors can complicate the selection of drug treatment. However, this study revealed that the differences in anti-HIV drugs selected for patients of different ages and sex are gradually diminishing. This result suggests that the standardization of drug treatment for HIV infection has progressed rapidly within the study period.

Limitations of this study include the following. First, it is not possible to know the actual usage of drugs that are not included in the NDB Open Data Japan. Because the NDB Open Data Japan provides data on the 100 most prescribed drugs in descending order; therefore, this study could not consider anti-HIV drugs that are not included in the top 100 list. In addition, drugs with a higher number of tablets taken per day may be ranked higher than drugs with a lower number. This can, in some cases, reverse the number of patients using an individual drug. Second, this study cannot consider the timing of cART initiation or the death of patients during cART. Therefore, the number of patients estimated is less than the actual number of patients undergoing cART.

## 5. Conclusions

This study evaluated the actual use of anti-HIV drugs in Japan from FY 2016 to FY 2019. Furthermore, for continuous survey on the actual use of anti-HIV drugs, the potential of epidemiological surveys using the NDB Open Data Japan was demonstrated. Researchers must choose the best research targets based on their own interests, and the NDB Open Data Japan is expected to be recognized as one of these targets and may be used frequently in future research.

## Figures and Tables

**Figure 1 ijerph-19-12130-f001:**
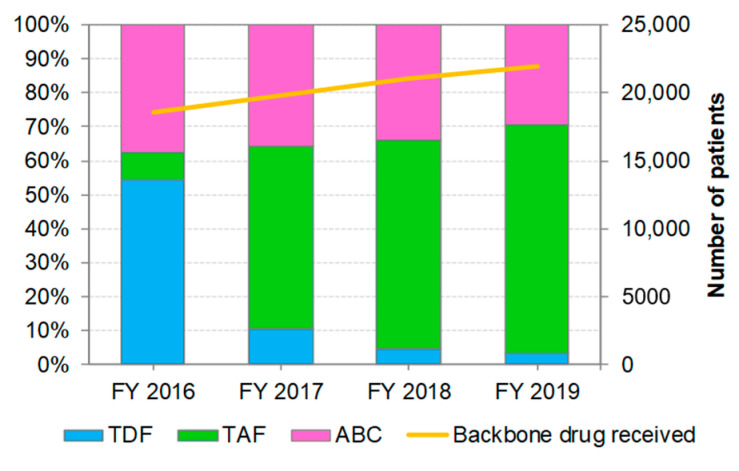
Changes in backbone drug prescriptions from FY 2016 to FY 2019. The yellow line indicates the number of patients estimated by the prescription volume of TDF, TAF, and ABC. The analysis was performed on drugs listed in Table 1. TDF: tenofovir disoproxil fumarate, TAF: tenofovir alafenamide fumarate, ABC: abacavir.

**Figure 4 ijerph-19-12130-f004:**
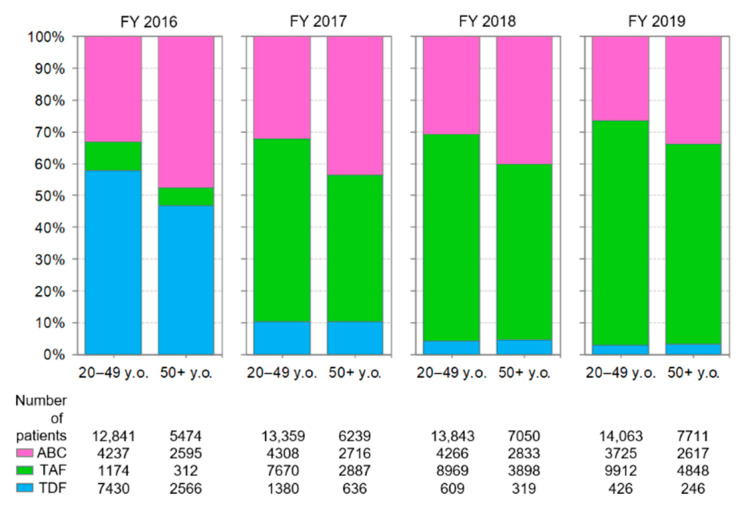
Changes in the prescriptions of backbone drugs by age group. The analysis was performed on drugs listed in Table 1. ABC: abacavir, TAF: tenofovir alafenamide fumarate, TDF: tenofovir disoproxil fumarate.

**Figure 5 ijerph-19-12130-f005:**
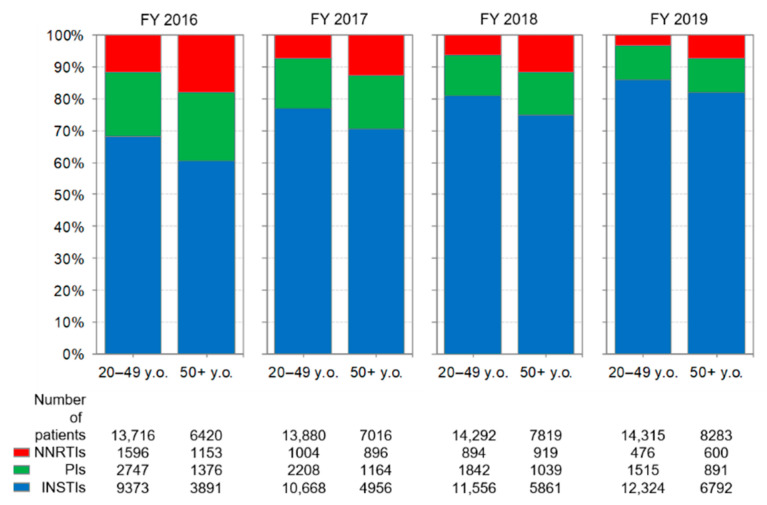
Changes in the prescriptions of anchor drugs by age group. The analysis was performed on drugs listed in Table 1. NNRTI: non-nucleoside reverse transcriptase inhibitor, PI: protease inhibitor, INSTI: integrase strand transfer inhibitor.

**Figure 6 ijerph-19-12130-f006:**
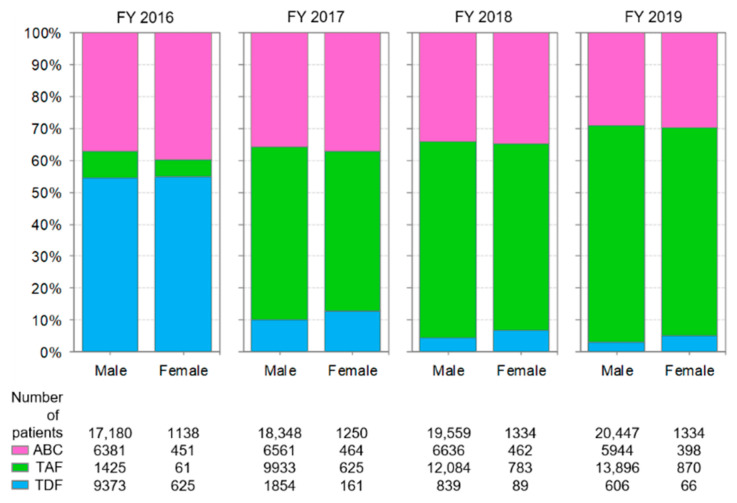
Changes in the prescriptions of backbone drugs by sex. The analysis was performed on drugs listed in Table 1. ABC: abacavir, TAF: tenofovir alafenamide fumarate, TDF: tenofovir disoproxil fumarate.

**Figure 7 ijerph-19-12130-f007:**
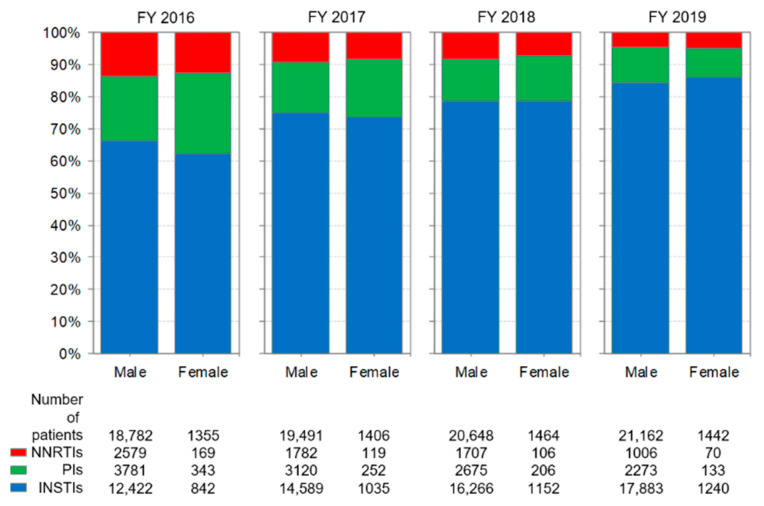
Changes in the prescriptions of anchor drugs by sex. The analysis was performed on drugs listed in Table 1. NNRTI: non-nucleoside reverse transcriptase inhibitor, PI: protease inhibitor, INSTI: integrase strand transfer inhibitor.

**Table 1 ijerph-19-12130-t001:** The list of anti-HIV drugs included in the 3rd, 4th, 5th, and 6th NDB Open Data Japan.

Product Name	Ingredients	Mechanism of Action			STR	NDB Open Data Japan			
		Anchor Drug	Backbone Drug	Others		3rd	4th	5th	6th
BIKTARVY^®^ Combination Tablets	BIC, TAF, FTC	INSTI	NRTI		Yes				〇
CELSENTRI^®^ Tablets 150 mg	MVC			CCR5 antagonist	No	〇	〇	〇	〇
Combivir^®^ Combination Tablets	AZT, 3TC		NRTI		No	〇	〇		
COMPLERA^®^ Combination Tablets	RPV, TDF, FTC	NNRTI	NRTI		Yes	〇			
Descovy^®^ Combination Tablets LT	TAF, FTC		NRTI		No		〇	〇	〇
Descovy^®^ Combination Tablets HT	TAF, FTC		NRTI		No	〇	〇	〇	〇
EDURANT^®^ Tablets 25 mg	RPV	NNRTI			No	〇	〇	〇	
Epivir^®^ Tablets 150 mg	3TC		NRTI		No	〇	〇	〇	〇
Epzicom^®^ Combination Tablets	ABC, 3TC		NRTI		No	〇	〇	〇	〇
Genvoya^®^ Combination Tablets	EVG/Cobi, TAF, FTC	INSTI	NRTI	Cobi: Booster	Yes	〇	〇	〇	〇
ISENTRESS^®^ Tablets 400 mg	RAL	INSTI			No	〇	〇	〇	〇
ISENTRESS^®^ Tablets 600 mg	RAL	INSTI			No			〇	〇
INTELENCE^®^ Tablets 100 mg	ETR	NNRTI			No	〇	〇	〇	〇
Kaletra^®^ Combination Tablets	LPV/RTV	PI		RTV: Booster	No	〇	〇	〇	〇
Kaletra^®^ Combination Oral Solution	LPV/RTV	PI		RTV: Booster	No	〇	〇	〇	〇
Lexiva^®^ Tablets 700 mg	FPV	PI			No	〇			
Norvir^®^ Tablets	RTV	PI		Booster	No	〇	〇	〇	〇
PREZCOBIX^®^ Combination Tablets	DRV/Cobi	PI		Cobi: Booster	No		〇	〇	〇
PREZISTA^®^ Tablets 600 mg	DRV	PI			No	〇	〇	〇	
PREZISTANAIVE^®^ Tablets 800 mg	DRV	PI			No	〇	〇	〇	〇
Retrovir^®^ Capsules 100 mg	AZT		NRTI		No	〇	〇	〇	〇
REYATAZ^®^ Capsules 150 mg	ATV	PI			No	〇	〇	〇	
STOCRIN^®^ Tablets 200 mg	EFV	NNRTI			No	〇	〇	〇	〇
STOCRIN^®^ Tablets 600 mg	EFV	NNRTI			No	〇	〇	〇	〇
Stribild^®^ Combination Tablets	EVG/Cobi, TDF, FTC	INSTI	NRTI	Cobi: Booster	Yes	〇			
SYMTUZA^®^ Combination Tablets	DRV/Cobi, TAF, FTC	PI	NRTI	Cobi: Booster	Yes				〇
Tivicay^®^ Tablets	DTG	INSTI			No	〇	〇	〇	〇
Triumeq^®^ Combination Tablets	DTG, ABC, 3TC	INSTI	NRTI		Yes	〇	〇	〇	〇
Truvada^®^ Combination Tablets	TDF, FTC		NRTI		No	〇	〇	〇	〇
VIRACEPT^®^ Tablets 250 mg	NFV	PI			No	〇	〇		
Viread^®^ Tablets 300 mg	TDF		NRTI		No	〇			
Ziagen^®^ Tablets 300 mg	ABC		NRTI		No	〇	〇	〇	〇

3TC: lamivudine, ABC: abacavir, ATV: atazanavir, AZT: zidovudine, BIC: bictegravir, Cobi: cobicistat, DRV: darunavir, DTG: dolutegravir, EFV: efavirenz, ETR: Etravirine, EVG: elvitegravir, FPV; fosamprenavir, FTC: emtricitabine, LPV: lopinavir, MVC: maraviroc, NFV: nelfinavir, RAL: raltegravir, RPV: rilpivirine, RTV: ritonavir, TAF: tenofovir alafenamide fumarate, TDF: tenofovir disoproxil fumarate, CCR5: C-C chemokine receptor type 5, INSTI: integrase strand transfer inhibitor, NRTI: nucleoside reverse transcriptase inhibitor, NNRTI: non-nucleoside reverse transcriptase inhibitor, PI: protease inhibitor, STR: single-tablet regimen.

## Data Availability

The 3rd, 4th, 5th, and 6th NDB Open Data Japan can be downloaded from the website of MHLW (https://www.mhlw.go.jp/stf/seisakunitsuite/bunya/0000177182.html, accessed on 31 October 2021).

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
