# Peer review of "Understanding the Actual Use of Anti-HIV Drugs in Japan from 2016 to 2019: Demonstrating Epidemiological Relevance of NDB Open Data Japan for Understanding Japanese Medical Care"

_ijerph, 2022, doi:10.3390/ijerph191912130_

Round 1

Reviewer 1 Report

The paper is clearly written, well composed, clearly presents the results. It is particularly easy to follow results because it has a sufficient number of clear tables. Investigation in this study answered his goal, which was already set in the title of this paper. The discussion explained  the results well and suggested a logical and useful continuation of the next research. It is about the use of STR cART in HIV patients with comorbidities that are discreetly mentioned in the discussion, eg in patients with kidney dysfunction or in special conditions such as pregnancy, but also  in a possible whole range of other comorbidities and conditions accompanying HIV/AIDS. The paper is supported by a sufficient number of references.

Reviewer 2 Report

Reviewer comment on Tanaka et al.: Understanding the actual use of anti-HIV drugs in Japan from 2016 to 2019: Demonstrating epidemiological relevance of NDB Open Data Japan for understanding Japanese medical care.

Dear Sirs,

easy-to-use and accurate resources for epidemiological analysis are a must in the understanding of trends, as well as in the reasonable management of national healthcare. Therefore, the NDB Open Data Japan seems an invaluable tool.

Meanwhile, I must highlight some minor and major points to review. First of all, the utility of the tool is undoubted, but the core limitation of the article, is that authors lost the aim while tried to show plausible results.

Abstract; Page 1; Line 10.: NDB seems an abbreviation, therefore in needs some explanation.

P 1; L 12.: The aim of the study needs revision, as the study itself rather focuses on utilization, and only discussion compares it with other studies.

P 1; L 20.: As I will mention it in the article, I do not see remarkable differences by age or sex, therefore the “gradually diminish” is not really accurate.

Introduction; P 2; L 82.: The article concentrates on the description of accessible data from NDB Open Data Japan. Results does not mention any comparison with source data, or other literature. Therefore, I would emphasize to rethink the aims or request access and ethical approval and do further research with source data, and compare the Open Data with it.

P 5; Table 1.: Please modify tables to fit the page, or manage the headers!

P 5; L 151. + Figure 1.: while it is noteworthy to not use unnecessary statistics (i. e. 4 data is a really low number), it is also better to avoid statistical terms, like “increased almost linearly”. This may be a bit misleading. It have to be also highlighted, that the first sight gives an impression of a remarked increment – meanwhile the real value must be lower, as the starting point of the graph is 16,000. Also, in the main text, at least two ways of calculation was mentioned for patients, who received backbone drugs, but on the figure it is not declared.

It is also misleading, that calculations were made by the three named drugs, and other backbone drugs were not taken in account for the 100%. It would be feasible to have a 4th block of drugs for other backbone drugs, if it is available, or calculate an approximate amount (maybe, by literature data).

P 7; Figure 3.: It would be preferred to describe, how to evaluate the lines of INSTI, TDF, TAF and ABC

P 8; L 227.: The authors say, write about slight differences between sexes from the beginning, therefore it is a bit misleading to write, there is no sex differences in the last year. (These statements may need statistical comparison as well.)

P 9; L 242.: From “The NDB includes… ”: information provided here about the NDB is better characterized, easier to understand and more concrete, therefore may be important for a standard reader from the beginning. Hence, it would be preferred to move to the introduction, since it better describes the relevance of NDB Open Data Japan.

P 9; L 256.: From “From NDB data…”: these comparisons were mentioned in the aims, so this should be in the results. Meanwhile, plagiarism should be avoided, therefore a new data-extraction and comparison would be preferred.

P 10; L 301.: Is there any calculation, wherewith the amount of not included drugs can be approached?

Reviewer 3 Report

This study presents a survey in the Japanese Health Database with the first objective to describe trends in HIV drug prescriptions and the second objective to explore the relevance of these types of surveys in this recent open database.

These are important issues to observe changes in prescriptions from an easily accessible database.

However, I do have several questions, comments, and suggestions.

My major concerns are:

-          drug-drug interactions are forgotten: the risk of DDI is considered for the choice of prescription. It deserves to be mentioned in the introduction l45 + l 56 (cf Demessine L, Peyro-Saint-Paul L, Gardner EM, Ghosn J, Parienti JJ. Risk and Cost Associated With Drug-Drug Interactions Among Aging HIV Patients Receiving Combined Antiretroviral Therapy in France. Open Forum Infect Dis. 2019 Mar 22;6(3):ofz051. doi: 10.1093/ofid/ofz051. PMID: 30949521; PMCID: PMC6440683.)

-          The discussion should be organized in the order of the objectives: first the trends of prescription, then the interest of this type of survey.

-          Perspectives of prescriptions surveys in HIV could be developed in discussion (what about biotherapies DOLU/3TC and DOLU/RILPI?)

-          Maybe compare with Europe and US?

Minor concerns:

-          L 18, Abstract: STRs in 2019 to compare with 2016

-          L50: References of guidelines

-          L62: how many subjects in the database?

-          L94: contradiction of “volume <1000” with l126

-          Table 1: Maj and min: harmonize the name of drugs

-          L171: “eventually”, not in results

-          L190: to compare with 2016

-          L221: sentence to suppress? (not in results)

-          L236: this study revealed the “evolutive” instead of “actual”?

-          L289: cite Parienti JJ on adherence

Looking forward to reading a new version of this study,

Best regards,

Round 2

Reviewer 2 Report

Reviewer comment on Tanaka et al.: Understanding the actual use of anti-HIV drugs in Japan from 2016 to 2019: Demonstrating epidemiological relevance of NDB Open Data Japan for understanding Japanese medical care. (Review 1)

I deeply appreciate the changes! For now, only one major change is needed, since Figure 7 is inaccurate, and may contain false information. This have to be changed, otherwise minor and linguistical comments are present.

Comment 1.

Answer is accepted, thank you.

Comment 2

The answer is accepted.

Comment 3.

Answer is accepted.

Minor comment (1).: It would be good to emphasize in the first few sentence, why it is important.

Comment 4.

I would advise minor revision (2).: While I also not prefer to use unnecessary statistical analysis, gradually diminish refers to a greater kind of difference in the starting point for a standard reader. Hence visual analysis seems appropriate, so, I rather thought about an explanation or comment like this one: "With the advent of newer drugs and regimens, the differences in anti-HIV 20 drugs prescribed to patients of different ages and sex gradually diminished; however, it was not remarkable in the first periods between groups, especially between males and females."

Comment 5

Answer is accepted.

Comment 6.

Answer is accepted.

Comment 7.

The answers and changes generally accepted.

Minor revision 3.: It would be less likely to be misleading, if every Y axis starts from 0.

Minor revision 4.: P 6; L 168.: As in the first sentence, the authors already mentioned Figure 1, and the paragraph has not changed topic, it is unnecessary to indicate here the Figure again.

Minor revision 5.: please delete the almost linearly from P 6 L 175 as well. (It won’t distort the sentence.)

Comment 8:

Answer is accepted.

Minor revision 6: In this case I would recommend to indicate in the description of every relevant Figures, that the analysis may have this limitation. (Something like: “The analysis has been performed with the top 100 drugs.” for these Figure.)

Comment 9.:

Answer is accepted.

Comment 10.

Answer is accepted.

Major revision 7.: Which diagram is for Figure 7? Why are there differences between the two diagrams?

Comment 11.:

Answer is accepted.

Minor revision: P 11; L 307.: "viological" should be corrected.

Comment 12.:

Answer accepted, minor revisions were advised at Comment 4.

Comment 13.:

Answer is accepted.

Author Response

添付ファイルをご覧ください。

Reviewer 3 Report

Thank you for the job!

Author Response

添付ファイルをご覧ください。
